# Considerations on the Kinetic Processes in the Preparation of Ternary Co-Amorphous Systems by Milling

**DOI:** 10.3390/pharmaceutics15010172

**Published:** 2023-01-03

**Authors:** Yixuan Wang, Thomas Rades, Holger Grohganz

**Affiliations:** Department of Pharmacy, Faculty of Health and Medical Sciences, University of Copenhagen, 2100 Copenhagen, Denmark

**Keywords:** polymer, ternary amorphous system, poorly water-soluble drugs, molecular interaction, phase separation, miscibility, Gordon-Taylor equation

## Abstract

In non-strongly interacting co-amorphous systems, addition of a polymer, to further stabilize the co-amorphous systems, may influence the phase behavior between the components. In this study, the evolution of the composition of the amorphous phase in the ternary system carvedilol (CAR)-tryptophan (TRP)-hydroxypropylmethyl cellulose (HPMC) was investigated, based upon previously formed and characterized binary systems to which the third component was added (CAR − TRP + HPMC, CAR − HPMC + TRP and TRP − HPMC + CAR). Ball milling was used as the preparation method for all binary and ternary systems. The influence of the milling time on the co-amorphous systems was monitored by DSC and XRPD. Addition of HPMC reduced the miscibility of CAR with TRP due to hydrogen bond formation between CAR and polymer. These bonds became dominant for the interaction pattern. In addition, when CAR or TRP exceeded the miscibility limit in HPMC, phase separation and eventually crystallization of CAR and TRP was observed. All ternary co-amorphous systems eventually reached the same composition, albeit following different paths depending on the initially used binary system.

## 1. Introduction

Solid dispersions are considered as an effective approach to address the low solubility, and thus potentially the bioavailability challenge, of many poorly-water soluble drugs [1]. In a solid dispersion, the drug can be dispersed in amorphous or crystalline form and it has been shown that the presence of the amorphous forms leads to a higher increase in solubility improvement compared with a crystalline form [2,3]. Polymeric solid dispersion form the bulk of studies performed in this area, whilst over the last decade co-amorphous systems also gain increasing interest. Co-amorphous systems are defined as amorphous systems consisting of one or more small molecular entities, usually a drug and a co-former [4,5]. Following am initial focus on drug-drug systems [6], amino acids became the center of attention as co-formers for co-amorphous systems [7,8]. The stabilization of co-amorphous systems usually occurs via molecular mixing or molecular interactions in the form of ionic bonds, hydrogen bonding or hydrophobic interaction. The majority of co-amorphous systems is prepared via ball milling, other frequently used methods include spray-drying or melt-quenching [9]. As a combination of polymeric amorphous solid dispersion and small molecule co-amorphous systems, ternary co-amorphous systems can be regarded as a promising strategy to design amorphous drug delivery formulations with high apparent water solubility as well as high physical stability of the drug [10,11]. In recent studies, polymer-containing ternary co-amorphous systems with improved solubility and physical stability have been developed. These systems were based on binary co-amorphous systems and polymers with a high glass transition temperature in order to strengthen intermolecular interactions and reduce molecular mobility of the drug [12,13,14,15]. However, new challenges have arisen, as additional interactions between the polymer and the drug can be formed in the ternary co-amorphous systems. Furthermore, recrystallization can be initiated due to limited miscibility between the components, and because polymers can change the mutual miscibility of the small molecules [16,17,18,19]. Many drugs have limited solubility in polymeric carriers, thus phase separation between the drug and polymer can also occur if the drug is initially supersaturated in the polymer [15]. This phase separation is usually followed by rapid nucleation and crystal growth of the drug [20].

Carvedilol (CAR) is a BCS class II drug with poor water solubility and is often used as a model drug [2,21,22]. Non-polar amino acids, specifically tryptophan, have been shown to be a good first choice as co-former for a given drug in a binary co-amorphous system [23]. Both ball milling for the preparation of co-amorphous systems and the CAR-TRP system as such have been investigated thoroughly earlier [24,25,26]. In a previous study, single-phase ternary co-amorphous carvedilol (CAR)-tryptophan (TRP)-polymer systems were prepared by ball milling and compared with the binary co-amorphous system CAR-TRP [2,21,22]. It could be shown that the addition of a small amount of polymer (10% *w*/*w* of the total solid content) to the binary systems improved the dissolution behavior of the drug by increasing the initial dissolution rate and maintaining supersaturation for a longer period of time.

However, the T_g_s of co-amorphous CAR-TRP-HPMC systems decreased during the ball milling process, and gradually tended to change towards the T_g_ of the CAR-HPMC binary system at the end of the milling process. Moreover, reflections between 17 to 19 °2θ gradually appeared in the XRPD diffractograms, corresponding to characteristic reflections of both CAR and TRP. DSC thermograms and XRPD diffractograms indicated that phase separation and recrystallization of CAR or TRP could occur during the ball milling process [18]. In the conversion of a binary system to a ternary system upon ball milling, different pathways might lead to different or similar end products. Therefore, it is necessary to investigate the influence of the polymer in ternary co-amorphous systems on the kinetic processes in the preparation of ternary co-amorphous systems by ball milling.

The objective of the current study was to determine the phase evolution and recrystallization of CAR or TRP in polymer-containing ternary co-amorphous systems upon increasing ball milling time. To obtain a mechanistic view on the kinetics, mixing of the three components was performed in different sequences. First binary systems were prepared (“A − B”) followed by the addition of the third component (“+ C”). Preparing three “A − B + C” co-amorphous systems, the feeding sequence is altered during the preparation process, thereby enabling a statement of the effect of the addition sequence of the three components on the molecular interaction pattern of the ternary co-amorphous system.

## 2. Materials and Methods

### 2.1. Materials

Carvedilol (CAR, MW = 406.47 g/mol) was purchased from Cipla Ltd. (Mumbai, India); L-tryptophan (TRP, MW = 204.23 g/mol) was used as a co-former and was purchased from Sigma-Aldrich (St. Louis, MO, USA). Hydroxypropylmethyl cellulose (HPMC, Pharmacoat 603, substitution type 2910, viscosity 3 mPas) was purchased from Shin-Etsu Chemicals Co. (Niigata, Japan).

### 2.2. Methods

#### 2.2.1. Ball Milling of Ternary Co-Amorphous Systems (A − B + C)

The ball-milled samples were prepared in a mixer mill MM 400 from Retsch GmbH & Co. (Haan, Germany). Stainless steel jars (25 mL) with two stainless steel balls (12 mm diameter) at a milling frequency of 30 Hz were used for the ball milling. To form three A − B + C combinations (CAR − TRP + HPMC, CAR − HPMC + TRP and TRP − HPMC + CAR), components A and B, which represent two of the three starting materials, were ball milled for 60 min to form a binary co-amorphous A − B system. Component C was added and the combined powder was milled for another 180 min. In order to reduce the likelihood for the absorption of moisture during ball milling, the jar was sealed with parafilm. The mixtures were measured by DSC and XRPD at predetermined time points during the milling process. In order to be consistent with our previous study [16], the molar ratio of CAR to TRP in the A − B + C system was maintained at 1:1 and the amount of HPMC was 10% (*w*/*w*) of the total solid content. The weight ratios of the components in the resulting A − B + C samples are shown in Table 1.

#### 2.2.2. Preparation of Physical Mixtures

In order to determine the origin of reflections found in the diffractograms, binary and ternary physical mixtures were prepared by manual mixing. A total mass of 2 g of drug and polymer was weighed into a mortar. After being mixed with a spatula, the powders were gently ground with a pestle for 60 s. The weight ratios of drug and polymer correspond to the respective co-amorphous systems (Table 2).

#### 2.2.3. Characterization of Solid-State Properties by X-ray Powder Diffraction (XRPD)

The solid-state properties of the samples were characterized using an X’Pert PANalytical PRO X-ray diffractometer (PANalytical, Almelo, The Netherlands), using Cu Ka radiation (1.54187 Å) and an acceleration voltage and current of 45 kV and 40 mA, respectively. The samples were placed on a sample holder plate and scanned from 5 to 35 °2θ in reflection mode, with a scan rate and scan step size of 0.0625 °2θ/s and 0.026 °2θ, respectively. Bragg-Brentano focusing geometry was used. The data was collected and analyzed using the software X’Pert Data Collector (PANalytical, Almelo, The Netherlands).

#### 2.2.4. Determination of the Glass Transition Temperature by Differential Scanning Calorimetry (DSC)

The T_g_ of the various samples was determined using a Discovery DSC (TA Instruments, New Castle, DE, USA) with a 50 mL/min nitrogen gas flow. A sample powder mass of 3–5 mg was accurately weighed into Tzero pans and sealed with hermetic lids with a pinhole. A modulated measuring mode was used for the DSC measurements. The samples were kept isothermal at −20.00 °C for 5 min before being heated up to 180.00 °C at an average rate of 3.00 °C/min, with an amplitude and a period of 0.2120 °C and 40 s, respectively. Data were collected and analyzed by the Trios software, version 5.0.0.44608 (TA Instruments, New Castle, DE, USA). The T_g_ was determined from the midpoint of the step change in the reversing heat flow signal. Recrystallization and melting events were detected from the total heat flow signal.

#### 2.2.5. Calculation of the Theoretical T_g_s Using the Gordon-Taylor Equation

Theoretical T_g_s were calculated using the Gordon-Taylor equation:
(1)$$Tg_{12}=\frac{\omega_{1}\times Tg_{1}+K\times \omega_{2}\times Tg_{2}}{\omega_{1}+K\times \omega_{2}}$$
where T_g12_ and T_gi_ are the T_g_s of the co-amorphous samples and the component i, respectively. ω_i_ is the mass fraction of the component i. K is a constant and can be calculated by the equation:
(2)$$K=\frac{Tg_{1}\times \rho_{1}}{Tg_{2}\times \rho_{2}}$$
where ρ_1_ and ρ_2_ are the densities of the two components. No information for the density of amorphous TRP was available in the literature, so it was calculated by the following equation:(3)$$\rho_{\mathrm{amorphous}\;\mathrm{TRP}}=\frac{\rho_{\mathrm{amorphous}\;\mathrm{CAR}}}{\rho_{\mathrm{crytalline}\;\mathrm{CAR}}}\times \rho_{\mathrm{crystalline}\;\mathrm{TRP}}$$
where ρ_crystalline CAR_ = 1.26 g/cm^3^ [27] and ρ_crystalline TRP_ = 1.30 g/cm^3^ [28]. The densities of amorphous CAR, amorphous TRP and HPMC are 1.24 [27], 1.282 and 1.285 [12] g/cm^3^, respectively.

#### 2.2.6. Investigation of Intermolecular Interactions by FT-IR

FT-IR spectra were collected using a Bomem FT-IR spectrometer (MB-Series, ABB Bomem Inc., Quebec, QC, Canada). Samples were scanned 64 times at a wavenumber range from 400 to 4000 cm^−1^ with a resolution of 4 cm^−1^. Data was collected by GRAMS/AI software (version 7.0, Thermo Fisher Scientific, Waltham, MA, USA) and analyzed using Origin software (version 9.6.0.172, OriginLab Corporation, Northampton, MA, USA). The reference compositions were prepared by ball milling for 180 min and are given in Table 3.

#### 2.2.7. Multivariate Data Analysis

The FT-IR spectral range from 800 to 1800 cm^−1^ and 3200 to 3500 cm^−1^ was used for further analysis. Pareto scaling transformation was conducted on the chosen range. The obtained preprocessed spectra were investigated by principal component analysis (PCA) using Simca 14.1 (Umetrics AB, Umeå, Sweden).

## 3. Results and Discussion

### 3.1. Characterisation of Binary CAR Systems

In order to evaluate the influence of adding a third component, the binary systems need to be understood first. CAR-TRP binary co-amorphous systems are described by the experimental Tgs (Figure 1a, DSC thermograms) and their comparison with the theoretical Tgs based on Gordon-Taylor equation (Figure 1b). Only one experimental Tg was found in each sample, which indicates the formation of homogenous co-amorphous systems. A melting peak of CAR was observed for CAR ratios of 60% (m/m) or higher. The change of the experimental Tg values as a function of CAR to TRP ratio follows the theoretical Tg values, calculated by the Gordon-Taylor equation. Thus, no specific or strong interactions between CAR and TRP in CAR-TRP co-amorphous systems could be detected.

In contrast, CAR-HPMC binary samples showed a negative deviation between the experimentally determined T_g_s and the theoretical T_g_s calculated by the Gordon-Taylor equation (Figure 1c,d), which indicates changes in the interactions pattern between HPMC and the drug in the CAR-HPMC binary systems [2,12,29]. The largest deviation could be found with concentrations between 30% to 40% CAR. A similar trend was observed for other systems and explained via the limited miscilibty of the two components, whereby an expected second T_g_ of a polymer-rich phase was not detected due to mixing of the domains upon heating; and the limited solubility of a compound in the polymer [30,31]. According to previous studies, the negative deviation indicates that the sum of interaction energy between drug and HPMC is lower than the sum of the drug-drug and HPMC-HPMC energies [32,33,34]. A modified approach to the Gordon-Taylor equation was used to investigate potential interactions between the CAR-HPMC system and an excess amorphous component. The CAR-HPMC ratio with the largest negative deviation from the theoretical values was treated as an individual component, and the excess CAR or HPMC was regarded as the second component in the calculation. The result of this calculation is shown as dashed line in Figure 1d. The experimentally determined T_g_ values closely follow the dashed lines, indicating that no obvious additional interactions occur between the CAR-HPMC system and the excess amorphous component.

In addition, according to the modified Gordon-Taylor equation, the theoretical T_g_s of the binary co-amorphous system CAR-TRP and CAR-HPMC at different composition ratios can also be determined. Therefore, it is possible to use the modified Gordon-Taylor equation to calculate the theoretical component ratios and describe the potential phase behavior of the A − B + C ternary co-amorphous systems. This will be discussed in more detail in Section 3.4.

### 3.2. Influence of Milling Time on the Ternary Systems

For all three A − B + C co-amorphous systems, the XRPD diffractograms and DSC thermograms showed an influence of the ball milling time on the phase behaviour of the ternary amorphous systems.

The XRPD diffractograms of the three A − B + C co-amorphous systems are shown in Figure 2a,c,e, respectively. Crystalline signals between 17 and 19 °2θ can be detected in all the three XRPD diffractograms after the addition of component C. These reflections might correspond to either CAR or TRP since the characteristic reflections of pure crystalline CAR and TRP can both be found in the area from 17 to 19 °2θ, respectively (18.5 and 18.7 °2θ, see Appendix A). When TRP was part of the original binary systems (Figure 2a,e), the crystalline reflection shows a tendency to increase with time, while a decrease is observed in the first 90 min followed by an increase until the endpoint (180 min of milling), when TRP is added into the binary system (Figure 2c).

The T_g_s of the three types of A − B + C amorphous systems after various milling times are shown in the DSC thermograms below (Figure 2b,d,e). The T_g_s are indicated by black arrows. The T_g_ of the CAR − TRP + HPMC system decreased from 70.9 °C to 42.1 °C with increasing ball-milling time (Figure 2b). The T_g_ of the CAR − HPMC + TRP system increased from 38.9 °C to 63.9 °C in the first 90 min of milling and then decreased to 42.6 °C after 180 min of milling (Figure 2d). The T_g_ of the TRP − HPMC + CAR system increased in the first 60 min to 71.2 °C and then decreased to 43.3 °C at the end of the ball milling process (Figure 2f). Simultaneously, with increasing milling time, the endothermic peak at about 115 °C gradually disappeared in the DSC thermographs [2]. This thermic event can be interpreted as a melting event of crystalline CAR (Appendix A).

The appearance of the crystalline reflections in the XRPD diffractograms of the A − B + C co-amorphous systems appears to be related to the changes in T_g_, with a tendentially stronger reflection in the XRPD at around 18 °2θ in systems with a lower glass transition. A rational for this finding might be the crystallisation of TRP and thus removal of the substance with the highest T_g_ from the ternary system, thereby leading to a decrease in the T_g_.

### 3.3. Identifying Intermolecular Interactions by Principal Component Analysis (PCA) of FT-IR Spectra

In order to provide further evidence for the findings from the thermal analysis, multivariate analysis of FTIR data was performed on each time point of the three A − B + C co-amorphous systems. The findings for the CAR − TRP + HPMC system will for the basis of the discussion, as corresponding results were seen for the other systems (Appendix A). According to previous research, a hydrogen bond has been proved to form between the N-H in CAR (3340 cm^−1^) and C-O in HPMC (1049 cm^−1^) in the ternary co-amorphous CAR-TRP-HPMC system [2]. Due to the small changes in the FTIR spectra with increasing milling time, PCA was chosen to investigate the pattern of the spectral variation. PCA was performed on the FT-IR spectra in the range from 800 to 1800 cm^−1^ and 3200 to 3500 cm^−1^. The FT-IR spectra of CAR-TRP, CAR-HPMC and TRP-HPMC binary systems and the CAR-TRP-HPMC ternary systems are used as used as control samples aiding the interpretation of the spectra of the A − B + C systems.

The spectral variation is reduced to a series of principal components (PCs) and summarized in the score plot. To understand the clustering in the score plots, the loading plots for A − B + C systems were examined. In the loading plots, positive and negative loading values correspond to the characteristic peaks from the respective FTIR spectra and were used to identify the components in the loading plots [12].

In a PCA model, the model quality can be explained based on parameters R^2^ (the explained variation, goodness of fit) and Q^2^ (goodness of prediction) [35,36]. The first principal component (PC-1) will always explain the largest part of the variation in the data matrix, compared to the succeeding PCs. However, this is not always this variation of interest, e.g., in the case of rather uniform samples [37]. Indeed, when looking at minor changes the information of interest can be presetned in the higher PCs. It is therefore of vital importance to compare the results from the PCA with meaningful exisiting knowledge, such as the shape of the original FTIR spectra. In the current investigation, the most obvious correlation between the raw spectra and the loadings was found for PC-2 and PC-4. Therefore, the score plots and loading plots of PC-4 versus PC-2 will be used in this part.

The score plot of CAR − TRP + HPMC system is shown in Figure 3a. With increasing ball milling time, the score value on PC-2 increases, whilst the score value on PC-4 decreases, indicating a change in the molecular interaction pattern, as also seen in the thermal analysis. In Figure 3b, the black arrow indicates characteristics peaks that can be matched with the corresponding peaks from the control samples. According to loadings, a positive loading value of PC-2 and PC-4 can be attributed to the interaction between CAR and HPMC, and CAR and TRP, respectively. Therefore, the decrease in PC-4 and increase in PC-2 score values, shows a decrease in the CAR-TRP interactions for the benefit of increasing CAR-HPMC interactions, thereby confirming the interpretation of the thermal analysis.

### 3.4. Development of Ternary Phase Diagrams

Based on the changing solid-state properties and the changes in the molecular interaction pattern described in Section 3.2 and Section 3.3, it can be assumed that three phases may exist in the A − B + C ternary system, caused by a limited solubility of both CAR and TRP in the polymer, as shown in Figure 4. One phase is a co-amorphous CAR-TRP-HPMC phase, while the other two phases are a separated CAR phase and a separated TRP phase; those phases might eventually either exist in amorphous or crystalline form. The appearance of a co-amorphous CAR-TRP phase, that could have been expected, based on the ability of the two compunds to form such a system, was ruled out, as this system would have resulted in a second T_g_, which was not observed. After the occurrence of above-mentioned phase separation, the A − B + C co-amorphous system at each measuring time point can be considered as “saturated”, and the solid-state forms of the ternary systems will be discussed in Section 3.4.1. Figure 4 also illustrates the basis for the calculation of the percentual composition of the amorphous systems in Section 3.4.2.

#### 3.4.1. Solid-State form in A − B + C Ternary Systems

HPMC is an amorphous material and previous studies have shown that HPMC in ternary systems will be involved in molecular interactions, but will not participate in the recrystallization process [38,39]. As the XRPD investigations (shown in Figure 2) were not conclusive in assigning the observed crystalline reflections to originate from CAR, TRP or both, a closer look at this phenomenon will be performed by thermoanalysis.

##### Investigation of the Solid-State form of TRP

Similar to the XRPD diffractograms of A − B + C systems (Figure 2a,c,e), the XRPD patterns of ball milled TRP also contain signals in the range 17 to 19 °2θ (Appendix A). In order to investigate the solid state behavior of TRP in the polymer, and exclude interference of CAR, a co-amorphous TRP-HPMC system was prepared (TRP + extra 10% HPMC). This system contained a TRP amount corresponding to the sum of CAR and TRP in the ternary systems, thus keeping the fraction of HPMC constant. The sample was ball milled for 120 min and then melted in the DSC. As expected, one endothermic peak (due to melting of crystalline TRP) appeared in the DSC thermogram (Figure 5a). Due to HPMCs amorphous nature, the endothermic peak in the DSC thermogram only represents the melting event of crystallized TRP. However, the detected T_m_ is lower than the literature value of pure TRP (286 °C) [40].

In order to confirm the existence of crystalline TRP in the solid dispersion, a physical mixture of TRP with additional 10% *w*/*w* HPMC was prepared by gentle grinding in a mortar, followed by melting in the DSC (TRP + extra 10% HPMC PM). As shown in Figure 5b, the DSC thermograms also showed one single endothermic peak, the onsets of the endotherm seen at 261.5 °C occurs at a similar temperature as the onsets of TRP + extra 10% HPMC in Figure 5a. The melting point of the physical mixture was about 4 °C higher compared with the ball milled samples. As the different preparation techniques can result in a different degree of dispersion, a difference in the melting points between the two techniques can be expected [41]. Combining the results of thermoanalytical testing and the XRPD diffractions, it can be concluded that TRP indeed has partly recrystallized upon ball milling.

##### Investigation of the Solid-State form of CAR

The occurrance of crystallised CAR is investigated by using the CAR − HPMC + TRP system as an example. This system has the highest T_g_ when ball milled for 90 min (Figure 2d). According to the previous findings, recrystallization began to occur after 90 min of ball milling. Figure 5c shows the DSC thermogram of the CAR-HPMC + TRP sample that was ball milled for 90 min.

In Figure 5c, four endothermic events can be observed: The first thermal event is between 40 and 65 °C. This event corresponds to the T_g_ (with enthalpic relaxation, data shown as total heat flow) of the CAR − HPMC + TRP ternary system, which is 63.9 °C as shown in the reversing heat flow signal in Figure 2d.The onset point of the second endothermic peak is 107.4 °C and the peak maximum is at 111.1 °C. This can be explained as a melting event of crystalline CAR. Since pure CAR will be fully amorphized after 60 min of milling, the occurrence of this event shows that recrystallization of CAR in the ternary system happens during the grinding process.The third thermal event at 147.3 °C corresponds to the T_g_ of TRP, which proves the separation of TRP from the ternary system during the grinding process.The starting point of the fourth endothermic peak is 231.1 °C and the endpoint is 256.0 °C. This event is attributed to the melting of TRP.

The occurrence of crystallized CAR can be verified by following an analogous approach as described in Section 3.4.1. The DSC heating curve of the CAR + TRP [1:1] mol/mol + extra 10% HPMC in Figure 5d shows two endothermic events. The first endotherm is seen between 100 to 130 °C and corresponds to the melting of pure CAR. The second endotherm is seen between 232.4 °C and 256.1 °C and corresponds to the 4th endothermic peak of the CAR − HPMC + TRP 90 min sample shown in Figure 5c, i.e., melting of TRP.

#### 3.4.2. Calculation of Phase Composition

Phase separation of CAR and TRP has been indicated in the ternary systems, and the separated CAR and separated TRP were able to recrystallize during the ball milling process. Depending on the stage of phase separation, the A − B + C ternary co-amorphous systems show different experimental T_g_s at each detection time according to the phase composition, i.e., the concentration of non-recrystallized compounds remaining with the polymer. The specific weight ratios of CAR to HPMC and CAR to TRP in the saturated co-amorphous CAR-TRP-HPMC phase can thus be calculated from the experimental T_g_ (Figure 2b,d,f) by the modified Gordon-Taylor equation (Figure 1b,d). Assuming that the T_g_ is indeed that of a ternary system, these specific weight ratios reflect the miscibility of CAR in HPMC and CAR in TRP in the saturated co-amorphous phase. Then, the concentration of CAR, HPMC and TRP in the saturated co-amorphous CAR-TRP-HPMC phase can be calculated from the miscibilities of CAR in HPMC and CAR in TRP mentioned above, provided that there are no specific interactions in the ternary systems that are not accounted for by interactions between the CAR-TRP and CAR-HPMC.

As shown in Figure 4, the miscibility of CAR in HPMC is [A%: 1 − A%] *w*/*w*, and the miscibility of CAR in TRP is [B%: 1 − B%] *w*/*w* in the saturated co-amorphous CAR-TRP-HPMC phase. The weight fractions of CAR, HPMC and TRP in the saturated co-amorphous CAR-TRP-HPMC ternary phase can be calculated as X, Y and Z as outlined in the equations below. These compositions were used in the development of the ternary phase diagrams (Figure 6).
(4)$$X=\frac{A\%}{A\%+\left(1-A\%\right)+\frac{1-B\%}{B\%}\times A\%}\;Y=\frac{\left(1-A\%\right)}{A\%+\left(1-A\%\right)+\frac{1-B\%}{B\%}\times A\%}\;Z=\frac{\frac{1-B\%}{B\%}\times A\%}{A\%+\left(1-A\%\right)+\frac{1-B\%}{B\%}\times A\%}$$

Figure 6 shows the kinetic evolution of the ternary amorphous system during ball milling. The starting point is the corresponding binary system. In all three systems, the trend goes from “more TRP” to “more HPMC”.

After milling for 180 min, the endpoints of the three ternary systems were located in the same area. This indicates that the component ratio the of ternary systems became similar upon prolonged milling, and that the feeding sequences did not affect the end result. The same final composition was achieved, independent of the starting binary systems, although via different routes.

A previous study has shown that the CAR-TRP hydrogen bond is weaker than the interaction between CAR and polymer [2]. Therefore, upon addition of component C, the A − B + C system firstly tends to form CAR-TRP hydrogen bonds and thereby shows the highest T_g_ at the solubility limit of TRP in CAR. After the system reaches the maximum concentration of TRP in CAR, the concentration of TRP in CAR gradually decreases due to the formation of CAR-HPMC hydrogen bonds upon further milling and eventually form the final composition of the saturated CAR-TRP-HPMC ternary system [42]. After 180 min of ball milling, all three investigated saturated CAR-TRP-HPMC amorphous systems show the lowest measured T_g_ at around 42.7 °C, which corresponds to the equilibrium composition of the miscibility gap. The decreasing T_g_ in the amorphous ternary systems upon milling is due to a minimum numbers of CAR-TRP bonds and the unorthodox behavior of the T_g_ development in HPMC systems described earlier. The average minimum solubility of CAR in TRP is CAR: TRP = [12.58 ± 1.41:1] _w/w_ = [6.32 ± 0.71:1] _mol/mol_ in the A − B + C ternary system with HPMC (illustrated in the phase diagrams by the tip of the red arrows in Figure 6). At the same time, the excess CAR and TRP will separate from the saturated CAR-TRP-HPMC ternary systems with the ratios of CAR:TRP = [1.51 ± 0.09:1] _w/w_ = [0. 76 ± 0.04:1] _mol/mol_.

## 4. Conclusions

In the ternary system CAR-TRP-HPMC, HPMC will provide a limited solubility for CAR and TRP. The leading role of the original CAR-TRP non-strong interaction for the systems was replaced by the increasing prevalence of hydrogen bonds between CAR and HPMC. The portion of CAR and TRP that exceed the solubility limit may recrystallize and gradually separate as two phases. The remainder of the compounds will form a saturated co-amorphous CAR-TRP-HPMC system. The same final composition was achieved, independent of the starting binary systems, although via different routes. In the design of ternary co-amorphous drug delivery systems, the effects of the polymer on the original molecular interactions between drug and co-former should thus be considered.

## Figures and Tables

**Figure 1 pharmaceutics-15-00172-f001:**
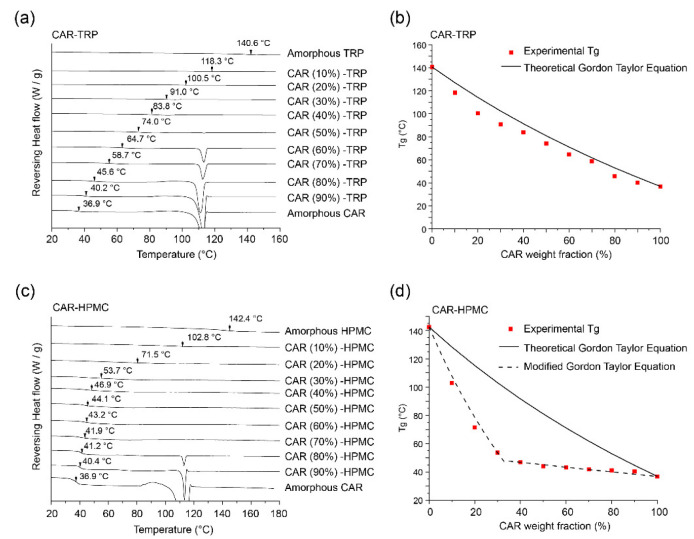
mDSC thermograms (reversing heat flow) of (**a**) co-amorphous CAR-TRP and (**c**) CAR- HPMC systems. The experimental T_g_s are indicated by arrows. (**b**) Comparison of experimental and theoretical T_g_ values of co-amorphous (**b**) CAR-TRP and (**d**) CAR- HPMC systems.

**Figure 2 pharmaceutics-15-00172-f002:**
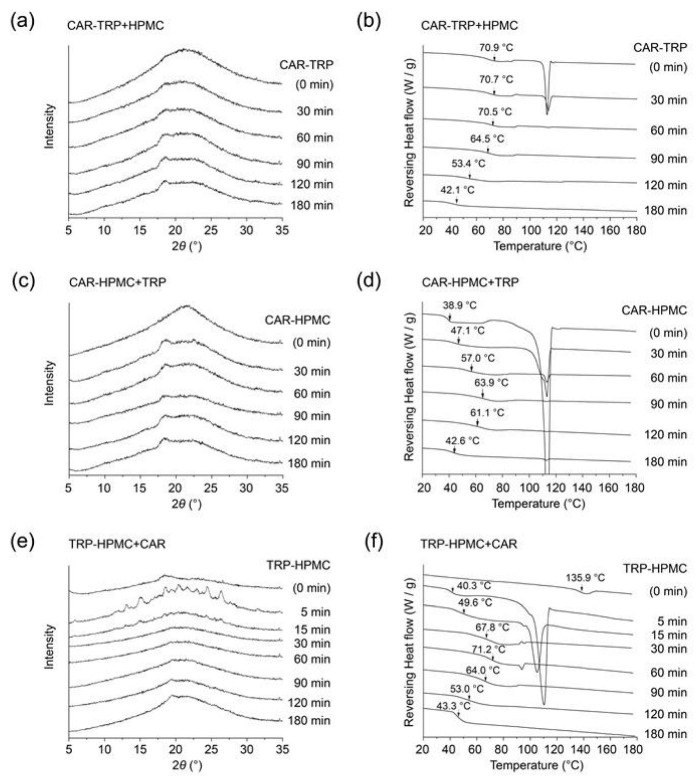
XRPD diffractograms and DSC thermograms of ball milled CAR − TRP + HPMC (**a**,**b**), CAR − HPMC + TRP (**c**,**d**) and TRP − HPMC + CAR (**e**,**f**) A − B + C amorphous systems. The T_g_s are indicated by arrows. The end point of ball milling of the “A − B” system is taken as the starting point of the milling of the “A − B + C” system.

**Figure 3 pharmaceutics-15-00172-f003:**
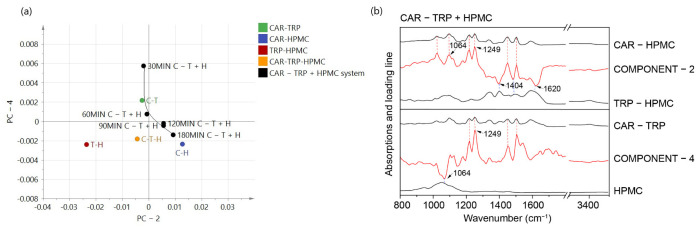
PCA score plot of FTIR spectra of CAR − TRP + HPMC, PC-2 was plotted against PC-4 (**a**). Loading plot of PC-2 and PC-4. Reference FTIR spectra of CAR-HPMC, TRP-HPMC, CAR-TRP binary amorphous systems and HPMC (**b**).

**Figure 4 pharmaceutics-15-00172-f004:**
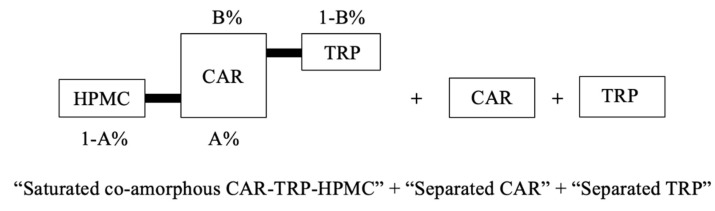
Schematic representation of phases in A − B + C ternary system.

**Figure 5 pharmaceutics-15-00172-f005:**
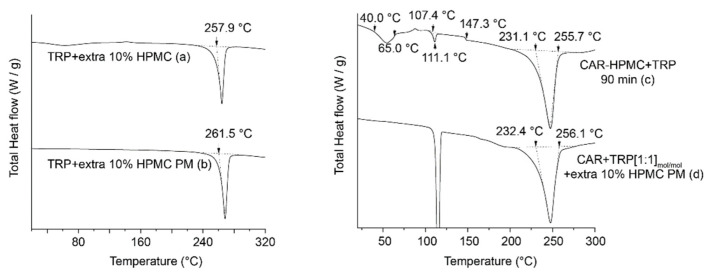
DSC thermograms of (**a**) TRP-HPMC at the milling endpoint; (**b**) TRP + extra 10% *w*/*w* HPMC physical mixture. The onset temperatures of the melting endotherm are indicated by arrows. DSC thermograms of (**c**) CAR-HPMC + TRP 90 min samples; (**d**) CAR + TRP [1:1] + extra 10% HPMC PM. The T_m_s and T_g_ are indicated by arrows.

**Figure 6 pharmaceutics-15-00172-f006:**
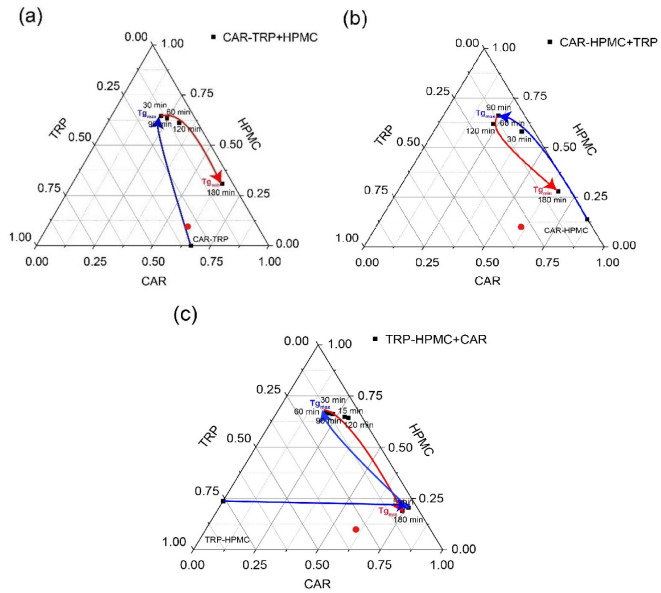
Ternary phase diagrams of CAR − TRP + HPMC (**a**), CAR − HPMC + TRP (**b**) and TRP − HPMC + CAR (**c**) A-B+C amorphous systems. The arrows indicate the change in the composition ratio of the saturated amorphous A − B + C ternary phase during the rise and fall of the T_g_ values upon milling. The positions of the maximum and minimum of the T_g_s in each ternary phase diagram are marked by T_g max_ and T_g min._ The theoretical composition without phase separaion is indicated with a red dot (CAR:TRP:HPMC = 60.5%:30.4%:9.1%).

**Table 1 pharmaceutics-15-00172-t001:** Weight ratios of the mixtures for the three A − B + C amorphous systems.

Samples	Component A − B Ratio (*w*/*w*)	Component C Ratio (*w*/*w*)
CAR − TRP + HPMC	CAR (60.5%) − TRP (30.4%)	HPMC (9.1%)
CAR − HPMC + TRP	CAR (60.5%) − HPMC (9.1%)	TRP (30.4%)
TRP − HPMC + CAR	TRP (30.4%) − HPMC (9.1%)	CAR (60.5%)

**Table 2 pharmaceutics-15-00172-t002:** Weight ratios of the binary and ternary physical mixtures.

Samples	CAR (*w*/*w*)	TRP (*w*/*w*)	HPMC (*w*/*w*)
TRP + extra 10% *w*/*w* HPMC PM	-	90.9%	9.1%
CAR + TRP [1:1] + extra 10% HPMC PM	60.5%	30.4%	9.1%

**Table 3 pharmaceutics-15-00172-t003:** Weight ratios of the binary and ternary co-amorphous reference systems for the FT-IR study.

Samples	CAR (*w*/*w*)	TRP (*w*/*w*)	HPMC (*w*/*w*)
CAR-TRP	66.6%	33.4%	-
CAR-HPMC	90.9%	-	9.1%
TRP-HPMC	-	90.9%	9.1%
CAR-TRP-HPMC	60.5%	30.4%	9.1%

## Data Availability

Data available on request as they do not have a public hub for file sharing at their University.

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
