# Peer review of "Considerations on the Kinetic Processes in the Preparation of Ternary Co-Amorphous Systems by Milling"

_pharmaceutics, 2023, doi:10.3390/pharmaceutics15010172_

Round 1

Reviewer 1 Report

1. One of the conclusions is that of CAR-TRP non-strong interaction being replaced by stronger CAR-HPMC interactions. There does not seem to be direct evidence other than inferences from thermal analysis in this manuscript. A more meaningful set of data would be from spectroscopic experiments of ternary system where the data is collected at the same timepoints as the PXRD samples. A binary system of CAR-HPMC or CAR-TRP independently may show evidence of this type of interaction, but it does not support how there is change in interaction as a function of milling time in a ternary system.

2. Another aspect that is highlighted is about the different phases as shown in schematic 3. Using thermal analysis for better understanding of phase separation in challenging as changes can happen during the heating and gathering information under similar conditions (for example room temperature) as PXRD would be more meaningful. In this aspect as well, performing spectroscopic analysis (Raman mapping, NMR etc) would really help differentiate domains and not be speculative in nature.

3. The authors mention appearance of crystalline peak around 18Ëš 2θ and changes to Tg as shown in figure 2 might be related to crystallization of TRP. Can the authors comment on why there is no increase in crystallinity as a function of time in the PXRD peaks beyond the appearance of that one peak? Additionally, is the CAR-TRP system truly co-amorphous at time zero based on figure 2a and b. It appears X-ray amorphous, but the melting endotherm of CAR could be due to presence of small seeds of crystalline CAR since no recrystallization exotherm is observed.

4. The authors use ball milling approach to generate amorphous material. What is the impact of ball milling on chemical stability of CAR or TRP and how the presence of any impurities it may be impacting the phase behavior? Also, there was no comment on how the moisture was controlled or was present and how it might have impacted the phase behavior.

Author Response

Reviewer 1:

Comment 1:

One of the conclusions is that of CAR-TRP non-strong interaction being replaced by stronger CAR-HPMC interactions. There does not seem to be direct evidence other than inferences from thermal analysis in this manuscript. A more meaningful set of data would be from spectroscopic experiments of ternary system where the data is collected at the same timepoints as the PXRD samples. A binary system of CAR-HPMC or CAR-TRP independently may show evidence of this type of interaction, but it does not support how there is change in interaction as a function of milling time in a ternary system.

Comment and action:

We have included FTIR data of the systems, presented in the form of a principal component analysis. The method section has been updated accordingly. The principal component analysis on the FT-IR of CAR-TRP+HPMC ternary system has been added in the manuscript as section 3.3. This one system has been chosen as it starts with the “originally investigated” CAR-TRP co-amorphous system. A corresponding picture was found for the other systems, which are included in the SI (Figure S2). The PCA provides evidence for the phenomenon shown in the thermal analysis with the CAR-TRP non-strong interaction being replaced by CAR-HPMC interaction.

Comment 2:

Another aspect that is highlighted is about the different phases as shown in schematic 3. Using thermal analysis for better understanding of phase separation in challenging as changes can happen during the heating and gathering information under similar conditions (for example room temperature) as PXRD would be more meaningful. In this aspect as well, performing spectroscopic analysis (Raman mapping, NMR etc.) would really help differentiate domains and not be speculative in nature.

Comment and action:

We agree that heating can change the sample, however, due to the amorphous nature, XRPD may not give a clearer picture. It is correct that we are looking at the bulk rather than the spatial distribution but for the kinetic evaluation this should be judged to be sufficient. We also consider that the inclusion of the FTIR data addresses this challenge.

Comment 3:

The authors mention appearance of crystalline peak around 18Ëš2θ and changes to Tg as shown in figure 2 might be related to crystallization of TRP. Can the authors comment on why there is no increase in crystallinity as a function of time in the PXRD peaks beyond the appearance of that one peak? Additionally, is the CAR-TRP system truly co-amorphous at time zero based on figure 2a and b. It appears X-ray amorphous, but the melting endotherm of CAR could be due to presence of small seeds of crystalline CAR since no recrystallization exotherm is observed.

Comment:

One reason for not seeing any other peak might be limited sensitivity of the XRPD. In addition, the peak around 18Ëš2θ is considered as the residual crystallinity of TRP, especially after a long-lasting mill (Kissi et al. Mol. Pharmaceutics 2018, 15 (9), 4247–4256).

With regard to CAR-TRP at time zero, previous studies showed that the CAR-TRP system could be fully amorphous after milling for 60 min. Some studies on the recrystallization of binary system with CAR also show that the recrystallization in DCS process is caused by excessive CAR rather than small seeds of crystalline CAR (Wang, Y. et al.,Int. J. Pharm. 2022, 617, 121625; EO Kissi et al. Mol. Pharmaceutics 2018, 15 (9), 4247–4256; Di et al. Molecules, 2021, 26(4): 801).

Comment 4:

The authors use ball milling approach to generate amorphous material. What is the impact of ball milling on chemical stability of CAR or TRP and how the presence of any impurities it may be impacting the phase behavior? Also, there was no comment on how the moisture was controlled or was present and how it might have impacted the phase behavior.

Comment and action:

Ball milling for the preparation of co-amorphous systems in general, and the CAR-TRP system as such have been investigated thoroughly earlier, the description and references have been added in the manuscript line 49. (Wang, Y. et al.,Int. J. Pharm. 2022, 617, 121625ï¼›Kasten et al., Eur. J. Pharm. Sci. 2016, 95, 28-35; Liu, J et al., Int. J. Pharm. 2020, 588, 119768.; Di et al. Molecules, 2021, 26(4): 801)

The following sentence was added into the method section: In order to reduce the likelihood for the absorption of moisture during ball milling, the jar was sealed with parafilm.

Furthermore, when collecting samples, the jars were kept in the cold room and opened after the jars cooled down to the temperature of the cold room. Heat energy, generated during ball milling, increases the temperature of the sample, and if not cooled, the temperature gap from sample to room temperature may cause the sample to become hygroscopic. Samples will then be stored in a -20 °C refrigerator and a dry environment with P2O5. These measures were also undertaken in earlier studies to avoid the absorption of moisture (Löbmann et al., European journal of pharmaceutics and biopharmaceutics, 2013, 85(3): 873-881.; Kasten et al., International journal of pharmaceutics, 2019, 557: 366-373.; Wu et al., Molecular pharmaceutics, 2019, 16(3): 1294-1304.).

Reviewer 2 Report

This manuscript demonstrates the complexity of co-amorphous formulations stabilized by polymers. Carvedilol and tryptophan along with HPMC are the model system, selected as it has been demonstrated that HPMC supplants the intermolecular interactions in the co-amorphous system. By forming ternary systems through ball-milling, it was shown that the equilibrium system is phase separated in to the ternary system as well as domains of each small molecule. This study will be of interest to the readers of Pharmaceutics.

Comments:

1.    The introduction needs to be expanded to provide sufficient background and context for the study. For example:

a.    A working definition of co-amorphous hasn’t been provided, as well as a summary of the state of the art around co-amorphous systems. Instead the first paragraph primarily jumps ahead assuming the reader is caught up on the literature, father than laying a foundation for the study ahead.

b.    Why is ball milling a suitable choice for preparation of co-amorphous systems? Why would it be selected over other possible choices?

2.    The type-setting on the rho letter in lines 124, 128 and possibly elsewhere needs to be fixed for publication. The character selected is hardly recognizable as the same one in the equation.

3.    Line 153: Percolation theory is not often used in amorphous research, and is not a concept most readers would be familiar with. I’d like to suggest that a simpler description (and additional reference) would be of tremendous value to your audience, as well as support your own conclusions: Miscibility is limited for the CAR-HPMC system. The same composition-based Tg trends were observed for indomethacin-HPMCAS and indomethacin-HPMC (https://pubs.acs.org/doi/full/10.1021/acs.molpharmaceut.6b00803), and the lack of a second Tg for the polymer-rich phase is suggested to be due to mixing of phase-separated domains during heating.

4.    Section 3.1: Why wasn’t TRP-HPMC binary system data studied? This seems essential to support the trends observed in the ternary system, and we’d also learn about the miscibility of TRP and polymer? A claim of limited solubility of TRP in polymer is made in line 207, without experimental support. (I just found Figure 4, which is more about investigating ball milling on these two components, rather than a compositional effect of the components like Figure 1).

5.    Figure 3:

a.    What about phases of CAR-TRP? TRP is supposed to be a suitable co-former for CAR, which is why these materials were selected for the study, so it seems reasonable that these materials may exist as a binary phase. If data to support that this wouldn’t be the case was found in your previous publication, it should be mentioned and referenced?

b.    Are the phases shown in the figure crystalline or amorphous, or could the be phases of both solid state forms?

c.     This figure would be clearer if presented in the same section as the equations, because the purpose and the function of the percentage notations is now more clear.

6.    Figure 4:

a.    I’m unclear how a conclusion of molecular-level mixing between two components can be assigned based on mortar/pestle mixing. Miscibility of components can be induced by heating in the DSC, which likely accounts for the observation of a reduced melting point (i.e. melting point depression). You also haven’t shown a TRP only scan up to the same temperatures (in Fig S1d, the highest temp shown is 180C).

b.    Can you write out the percentages of TRP and HPMC in these experiments? This level of detail was provided in methods for other experiments, and listing composition values as “extra” results is confusing.

7.    A homogenous composition value should be added to the ternary phase diagrams, so it’s clear how the equilibrium 180 minute value deviates from the theoretical composition.

8.    This study seems to be missing measurements of a ternary control composition. Could a ternary system be made by a solvent-mediated method? This would assist with the interpretation of crystallization in DSC curves.

Author Response

Reviewer 2:

This manuscript demonstrates the complexity of co-amorphous formulations stabilized by polymers. Carvedilol and tryptophan along with HPMC are the model system, selected as it has been demonstrated that HPMC supplants the intermolecular interactions in the co-amorphous system. By forming ternary systems through ball-milling, it was shown that the equilibrium system is phase separated into the ternary system as well as domains of each small molecule. This study will be of interest to the readers of Pharmaceutics.

Response:

Thank you very much for giving comments and suggestions to our article, and for the evaluation that the work will be of interest to the readers. In our revised manuscript, we improved the readability and content according to the suggestions of the reviewer.

Comment 1:

The introduction needs to be expanded to provide sufficient background and context for the study. For example:

  1. A working definition of co-amorphous hasn’t been provided, as well as a summary of the state of the art around co-amorphous systems. Instead, the first paragraph primarily jumps ahead assuming the reader is caught up on the literature, father than laying a foundation for the study ahead.
  2. Why is ball milling a suitable choice for preparation of co-amorphous systems? Why would it be selected over other possible choices?

Comment and action:

We agree with the reviewer that the introduction was rather short and maybe started a bit abruptly.  We have therefore increased the introduction to some degree. However, as co-amorphous systems as such are not that novel anymore, and ball milling also being the most common production method, we also did not want to overdo it.

Comment 2:

The typesetting on the rho letter in lines 124, 128 and possibly elsewhere needs to be fixed for publication. The character selected is hardly recognizable as the same one in the equation is hardly recognizable as the same one in the equation.

Action:

All the characters in the equation have been fixed.

Comment 3:

Line 153: Percolation theory is not often used in amorphous research and is not a concept most readers would be familiar with. I’d like to suggest that a simpler description (and additional reference) would be of tremendous value to your audience, as well as support your own conclusions: Miscibility is limited for the CAR-HPMC system. The same composition-based Tg trends were observed for indomethacin-HPMCAS and indomethacin-HPMC (https://pubs.acs.org/doi/full/10.1021/acs.molpharmaceut.6b00803), and the lack of a second Tg for the polymer-rich phase is suggested to be due to mixing of phase-separated domains during heating.

Comment and action:

Thank you very much for your suggestions. The explanation and suggested article (Ref 27) have been added.

Comment 4:

Section 3.1: Why wasn’t TRP-HPMC binary system data studied? This seems essential to support the trends observed in the ternary system, and we’d also learn about the miscibility of TRP and polymer? A claim of limited solubility of TRP in polymer is made inline 207, without experimental support. (I just found Figure 4, which is more about investigating ball milling on these two components, rather than a compositional effect of the components like Figure 1).

Comment and action:

The focus on the article was on amorphous systems including the drug CAR and did therefore not focus on the relationship between the two co-formers. The included principal component analysis on the FT-IR of CAR-TRP+HPMC ternary systems in section 3.3 shows the changing trend of the interaction from CAR-TRP to CAR-HPMC without TRP-HPMC becoming relevant. However, we also added the FTIR data to the supplementary information.

The limited solubility of TRP in polymer is also shown via the recrystallization of TRP (in section 3.4.1.1). Previous studies have also shown limited miscibility of drug in HPMC causing phase separation and recrystallization. (Lu, Q.; Zografi, G. Res. 1998, 15, 1202-1206.; Rask M B, et al. International Journal of Pharmaceutics, 2018, 540(1-2): 98-105.)

Comment 5:

Figure 3:

  1. What about phases of CAR-TRP? TRP is supposed to be a suitable co-former for CAR, which is why these materials were selected for the study, so it seems reasonable that these materials may exist as a binary phase. If data to support that this wouldn’t be the case was found in your previous publication, it should be mentioned and referenced?

Comment:

We would like to thank the reviewer for this comment and ensure that we also considered the possibility of a CAR-TRP phase, as CAR-TRP was chosen due to its potential to form a co-amorphous system. However, based on the reasons below, we ruled its appearance out in the presence of polymer.

The presence of a crystalline CAR-TRP phase, would have resulted in a co-crystal. However, this co-crystal has never been observed, presumably due to the likelihood of forming a co-amorphous system; and we have also not observed any novel diffractions in the XRPD.

The presence of CAR-TRP as a second independent amorphous phase should have resulted in a second Tg, which was also not observed. This is due to the shown mechanism of CAR “leaving TRP and moving to HPMC”.

The existence of a CAR-TRP phase could be debated for systems with very tiny amounts of polymer. In our previous systems the solid state of these systems was not investigated in detail. In stability experiments the presence of 1% HPMC with CAR-TRP led to the same stability as the CAR-TRP system (longer than 24weeks) whilst the CAR-HPMC recrystallized after 6 weeks. This would indirectly point to the existence of a CAR-TRP phase, however, it might also be due to the higher Tg of the ternary system compared with the binary CAR-HMPC system.

We have added a comment on this in connection with the figure.

  1. Are the phases shown in the figure crystalline or amorphous, or could the be phases of both solid-state forms?

Comment and action:

The ternary system CAR-TRP-HPMC is amorphous as has been pointed out in the figure.

The CAR and TRP phases can be amorphous or crystalline. The CAR or TRP beyond miscibility separate s single phase and gradually re-crystalized. In order to make the reader free from subjective judgment at the beginning of the discussion, it is not indicated that it is ‘re-crystallized CAR’ and ‘re-crystallized TRP’ but replaced by ‘separated CAR’ and ‘separated TRP’. It was added to the manuscript that the separated phases might eb both crystalline or amorphous.

  1. This figure would be clearer if presented in the same section as the equations, because the purpose and the function of the percentage notations is now clearer.

Comment and action:

We agree with the reviewer that there is a clear relationship between Figure 3 and the %-equations. However, the figure is also used to illustrate the basis for the description of the assumed systems discussed in chapter 3.4. and provides an overview of the entire Section 3.4. It would therefore not be appropriate to move the Figure3 to the calculation section.

We have added a comment in connection with Figure 3 to make the connection with the calculations clearer.

Comment 6:

Figure 4:

  1. I’m unclear how a conclusion of molecular-level mixing between two components can be assigned based on mortar/pestle mixing. Miscibility of components can be induced by heating in the DSC, which likely accounts for the observation of a reduced melting point (i.e. melting point depression). You also haven’t shown a TRP only can up to the same temperatures (in Fig S1d, the highest temp shown is 180C).

Comment:

We agree that this was not clear formulated and removed the phrase molecular mixing at one occasion as it was misleading. The mortar and pestle was used to achieve a physical mixture. The physical mixture was then heated for for DSC study. This procedure followed several references, which also clarify that no intermolecular interaction would form after mixing two components (Wu et al. Molecular pharmaceutics, 2019, 16(3): 1294-1304; Kissi et al. Pharmaceutics, 2019, 11(12): 6280; Löbmann et al. European Journal of Pharmaceutics and Biopharmaceutics, 2012, 81(1): 159-169.).

  1. Can you write out the percentages of TRP and HPMC in these experiments? This level of detail was provided in methods for other experiments and listing composition values as “extra” results is confusing.

Comment:

The percentage of each component is shown Table 1 and Table 2.

Comment 7:

A homogenous composition value should be added to the ternary phase diagrams, so it’s clear how the equilibrium 180-minute value deviates from the theoretical composition.

Comment and action:

The theoretical composition point in red has been added in Figure 6 which represents the component ratio if phase separation does not occur (CAR: TRP: HPMC= 60.5% : 30.4% : 9.1%). 

Comment 8:

This study seems to be missing measurements of a ternary control composition. Could a ternary system be made by a solvent-mediated method? This would assist with the interpretation of crystallization in DSC curves.

Comment:

The study is based on the formation of a binary system to which then a third component is added in order to investigate the kinetic pathway of the formation of the final system. As such, a ternary system cannot function as a starting system for the same methodology.

This idea is however investigated in an on-going project which uses different ternary mixtures as starting material. We also believe that a ternary system that has been produced via a very different method, such as spray-drying, would not be comparable with the systems of the current study

Reviewer 3 Report

Wang and co-workers characterize in their submitted study a ball-milling based approach to generate co-amorphous ternary formulations containing two APIs and the polymer HPMC. The evolution of the ternary composition and the mixing of the components in each other was investigated time-dependently via determination of the Tg and observation of melting peaks in a DSC. The three components were mixed and processed in different feeding sequences and finally yielded comparable ternary co-amorphous mixtures via different routes. It I a nice study that illustrates how the miscibility behavior can be applied to generate ternary co-amorphous formulations via a milling process. My comments are below, which should be considered prior to publication:

-          P11 line 329 ‘which corresponds to the lowest miscibility of CAR in TRP’ is misleading – this corresponds to the equilibrium composition of the miscibility gap, this composition is approximately pure CAR/TRP

-          It would be helpful having at least schematical phase diagrams that illustrate the expected equilibria in the ternary mixtures. Also Lines of constant Tg in a ternary diagram would be helpful (e.g. calculated via Gordon-Taylor, three glass-transition temperatures of relevance). I assume that the phase diagram possibly looks like the schematic figure illustrated below. Based on this, it can be seen that a certain Tg might be attributed to several compositions, and the observed Tgs are an indirect measure of the compositions.

-          Phrases like ‘CAR-TRP interaction being replaced by newly formed, stronger hydrogen bonds’ (e.g. Abstract) – should be revised in a way that the interactions are not really replaced, but additionally introduced, strong hydrogen bonds play an increasing role in the interaction profiles.

-          P5 (first lines) and Figure 1d– The calculation of the Tg for the demixed CAR/HPMC mixtures should be revised – The low, almost constant wCAR-independent Tg is that of the CAR-rich amorphous phase, which does hardly change with changing composition.

-          Was the Tg determined during the first heating ramp? It makes sense to determine this value for investigating the ball-milling process. I propose a heat-cool-heat ramp and check for some samples deviations to the second heating.

-          Figure 1d – this behavior is typical for immiscible API/polymer blends and the flat Tg course is expected to occur due to the immiscibility of CAR and HPMC (precipitation of amorphous CAR; compare for example Huang and Dai:  https://www.sciencedirect.com/science/article/pii/S2211383513000968).

-          Was a second Tg observed in some samples?

Author Response

Reviewer 3:

Wang and co-workers characterize in their submitted study a ball-milling based approach to generate co-amorphous ternary formulations containing two APIs and the polymer HPMC. The evolution of the ternary composition and the mixing of the components in each other was investigated time-dependently via determination of the Tg and observation of melting peaks in a DSC. The three components were mixed and processed indifferent feeding sequences and finally yielded comparable ternary-amorphous mixtures via different routes. It I a nice study that illustrates how the miscibility behavior can be applied to generate ternary co-amorphous formulations via a milling process. My comments are below, which should be considered prior to publication:

Comment:

We would like to thank the reviewer for the affirmation of our experimental strategy. We also thank the reviewer for the constructive questions and suggestions for this manuscript, the response of each question can be found below.

Comment 1:

P11 line 329 ‘which corresponds to the lowest miscibility of CAR in TRP’ is misleading – this corresponds to the equilibrium composition of the miscibility gap, this composition is approximately pure CAR/TRP

Action:

Changed according to the reviewer.

Comment 2:

It would be helpful having at least schematical phase diagrams that illustrate the expected equilibria in the ternary mixtures. Also Lines of constant Tg in a ternary diagram would be helpful (e.g. Calculated via Gordon-Taylor, three glass-transition temperatures of relevance). I assume that the phase diagram possibly looks like the schematic figure illustrated below. Based on this, it can be seen that a certain Tg might be attributed to several compositions, and the observed Tgs are an indirect measure of the compositions.

Comment:

We agree that in theory a certain Tg might be attributed to several compositions, considering the presence of 3 amorphous compounds. However when following the schematic from Figure 4 and the subsequent calculations one defined composition is allocated. Therefore lines of constant Tg would indicated the presence of compositions that would not follow the assumptions made in Figure 4, that form the basis of the manuscript. ‘The schematic figure illustrated below’ was not attached in the review report.

Comment 3:

Phrases like ‘CAR-TRP interaction being replaced by newly formed, stronger hydrogen bonds’ (e.g. Abstract) – should be revised in a way that the interactions are not really replaced, but additionally introduced, strong hydrogen bonds play an increasing role in the interaction profiles.

Comment and action:

According to the reviewer’s comments, these phrases have been reformulated throughout the manuscript.

Comment 4:

 P5 (first lines) and Figure 1d– The calculation of the Tg for the demixed CAR/HPMC mixtures should be revised – The low, almost constant w CAR-independent Tg is that of the CAR-rich amorphous phase, which does hardly change with changing composition.

Comment:

We agree with the reviewer that the slope of the “right arm” in CAR-HPMC appears rather low, a behavior that has often been spotted for HPMC behavior in ASDs. However, this impression is enhanced by the scaling with the extremely high Tg of pure HPMC. When comparing the systems from 30 % CAR to pure CAR, a decrease in Tg of 16.8 degrees is seen (Figure 1c). We deem the absolute difference to be sufficient for the calculative purposes performed. The methodology of using the modified Gordon-Tylor equation to calculate the Tg of a CAR-HPMC binary system with different component ratio has also been used earlier (Wang et al., International Journal of Pharmaceutics, 2022, 617: 121625.).

Comment 5:

Was the Tg determined during the first heating ramp? It makes sense to determine this value for investigating the ball-milling process. I propose a heat-cool-heat ramp and check for some samples deviations to the second heating.

Comment:

Corresponding to previous studies on co-amorphous systems with CAR, all the Tgs were determined during the first heating ramp (Wang et al., International Journal of Pharmaceutics, 2022, 617: 121625; Di et al. Molecules, 2021, 26(4): 801; Liu et al., International Journal of Pharmaceutics, 2020, 588: 119768.).

Comment 6:

Figure 1d – this behavior is typical for immiscible API/polymer blends and the flat Tg course is expected to occur due to the immiscibility of CAR and HPMC (precipitation of amorphous CAR; compare for example Huang and Dai: https://www.sciencedirect.com/science/article/pii/S2211383513000968).

Comment and action:

We would to thank the reviewer, the article is a good support for the behaviour in CAR-HPMC system. This article has been added as reference 28.

Comment 7:

Was a second Tg observed in some samples?

Comment:

All samples but one resulted in a single Tg. The only sample in the whole set-up where two Tgs were observed, was based on milling TRP (10% w/w)- HPMC for 120min. However as the second Tg of 162.21°C is above the Tgs of the starting compounds, indicating a potential artefact. As the sample also did not contain any CAR, this was disregarded for the article.

Round 2

Reviewer 1 Report

No additional comments.